# Approximating Language Model Training Data from Weights

**John X. Morris, Junjie Oscar Yin, Woojeong Kim, Vitaly Shmatikov, Alexander M. Rush**
Department of Computer Science
Cornell University
New York, NY
jxm3@cornell.edu

## Abstract

Modern language models often have open weights but closed training data. We formalize the problem of data approximation from model weights and propose several baselines and metrics. We develop a gradient-based approach that selects the highest-matching data from a large public text corpus and show its effectiveness at recovering useful data given only weights of the original and finetuned models. Even when none of the true training data is known, our method is able to locate a small subset of public Web documents can be used to train a model to close to the original model performance given models trained for both classification and supervised-finetuning. On the AG News classification task, our method improves performance from 65% (using randomly selected data) to 80%, approaching the expert benchmark of 88%. When applied to a model trained with SFT on MSMARCO web documents, our method reduces perplexity from 3.3 to 2.3, compared to an expert LLAMA model's perplexity of 2.0.

## 1 Introduction

Modern language models have been scaled up to 1 trillion parameters, which take 2 TB of disk storage to store in half-bit precision. Yet these models are often finetuned on thousands to millions of samples, the totality of which may take less than 1 GB of disk space. Although models may lack capacity to memorize their entire pretraining data (Morris et al., 2025), they can easily memorize smaller finetuning datasets.

In addition, it is common for language models to be *open-weights* but not *open-data*: their creators release the weights of the language model so that they can be run on local hardware, but do not release the datasets the models were trained on. For example, recent popular reasoning model DeepSeek-R1 (DeepSeek-AI, 2025) is trained from DeepSeek-Base (Liu et al., 2024), both of which are open-weights; however, there has been much public speculation over what data was used to train R1 from Base. Similarly, the instruction-tuned and base versions of the LLAMA-3 models (Meta-AI, 2024) are open-weights, but the instruction-tuning data is unknown.

If the weights are public, they may reveal some information about the training data. Several works in computer vision have approached the problem of reconstructing training data from model weights (Balle et al., 2022; Haim et al., 2022), or distilling data using a "trajectory" of multiple model checkpoints (Cazenavette et al., 2022; Guo et al., 2024). However, all these approaches require backpropagating gradients from the network output to its input, which is not possible with discrete language data.

Instead of generating data from scratch, we consider whether an adversary might *select* data points from a web-scale corpus to simulate the performance of the true finetuning data. This problem resembles conventional core-set selection, but is much harder: we have access to only a single final model and no validation data. These restrictions prevent us from using conventional approaches such as gradient matching (Xia et al., 2024).

We consider the additional fact that many open-weights models are fine-tunes of base models. A modern adversary has access to *two* model checkpoints: the original ("base")

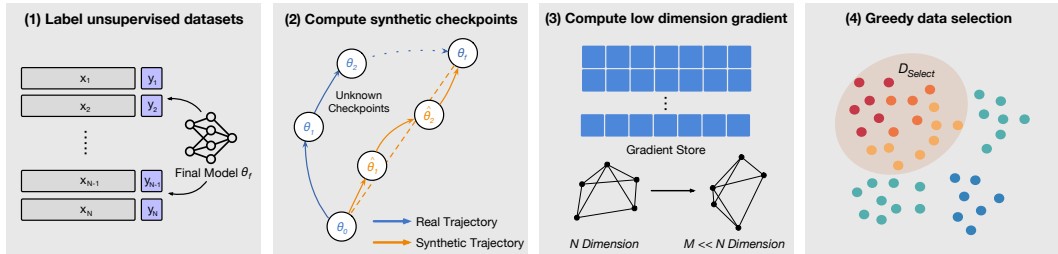

Figure 1: **SELECT Method Overview.** We approximate training data from initial and final language model checkpoints. Given an unsupervised seed dataset, we first use the final model $\theta_f$ to assign a label to each instance. To compute gradients efficiently, we leverage random projections under the Johnson-Lindenstrauss lemma, which guarantees that gradient projection preserves inner-product information. Finally, we perform greedy data selection by picking instances with the highest running gradient sum.

and fine-tuned ("final") model. We develop a new method (SELECT) that relies only on the signal from these two models: gradients from the base model and pseudolabels from the final model.

Our algorithm greedily selects datapoints from a seed set such that their sum points along the direction between the two models in weight space. **Gradient-based selection significantly outperforms all baselines across a variety of settings, showing that we are recovering essential information about the actual finetuning data from the weights of the fine-tuned model.** In the case of both text classification and supervised finetuning, even when data selected by our method is completely disjoint from the true finetuning set, our selected data is still of sufficiently high quality that a (different) model trained on this data achieves high accuracy on the downstream task. On the AG News classification task, our method improves performance from 65% (using randomly selected data) to 80%, approaching the expert benchmark of 88%. When applied to a model trained with SFT on MSMARCO web documents, our method reduces perplexity from 3.3 to 2.3, compared to an expert LLAMA model's perplexity of 2.0.

To measure the extent to which data selected by our method is similar to the true finetuning data, we compute metrics based on Optimal Transport (OT) of sentence embeddings between the true and selected datasets. A scaling analysis shows that with a larger "seed" dataset, our method outperforms the baselines according to this OT-based metric.

To run our method at scale, we propose efficiency improvements that make our gradient-based approach tractable even when searching over a set of millions of documents. Our method uses only the final-layer gradient of the model, which is relatively cheap to compute and known to be recoverable from any model that provides open-source access (Morris et al., 2023; Finlayson et al., 2024; Carlini et al., 2024). We show our method is effective at recovering useful data from models trained for both classification and supervised fine-tuning.[1]

## 2 Related Work

**Coreset selection.** Coreset selection aims to select a subset of training data such that models achieve performance comparable to training on the full dataset (John & Draper, 1975; Borsos et al., 2020). One line of methods develops utility functions to score and select data based on various features including n-grams, distances to k-means clusters, and model losses (Xie et al., 2023; Chen et al., 2012; Feng et al., 2022). Another line is optimization-based, with methods like Mirzasoleiman et al. (2020) selecting coresets whose gradients closely match those computed over the entire dataset. This approach has been extended by

---

[1]Our project is fully open-source. All code is available on Github.

Killamsetty et al. (2021b;a) to handle noisy and imbalanced datasets by performing gradient matching against a held-out validation set.

In language modeling, gradient-based coreset selection has proven valuable, with Pruthi et al. (2020) using estimated gradient information to identify mislabeled data and Xia et al. (2024); Lee et al. (2023) applying gradient matching in supervised fine-tuning to improve data efficiency. In contrast, our work focuses on recovering finetuning data from sparse training checkpoints, operating under the constraint of having access only to two model checkpoints, without any validation data.

**Dataset distillation.** Data distillation synthesizes a small synthetic dataset rather than selecting from existing data. Wang et al. (2020) first proposed data distillation as a bi-level optimization problem: the distilled dataset is treated as the inner optimization variable, while the model weights are treated as a function of this dataset in the outer optimization. Despite its theoretical elegance, this bi-level meta-learning framework incurs substantial computational costs. Follow-up works incorporate gradient matching as a supervised signal, aligning model's gradients trained on original and distilled datasets (Zhao et al., 2021; Lee et al., 2023). Further works have develop techniques like trajectory matching (Cazenavette et al., 2022; Cui et al., 2023; Yin et al., 2024), difficulty alignment (Guo et al., 2024), and kernel methods (Zhou et al., 2022). Our work extends trajectory matching methods and reformulates them for data selection in language modeling.

**Training data reconstruction and extraction.** A small number of works have attempted to directly recreate data from the weights of feed-forward neural networks (Haim et al., 2022; Buzaglo et al., 2023; Runkel et al., 2024) and convolutional neural networks (Balle et al., 2022). In the realm of language models, research has instead focused on the *extraction* of single training examples by direct generation from the models (Carlini et al., 2021; Nasr et al., 2023). Maekawa et al. (2024) and Nguyen et al. (2025) recently propose methods for generating synthetic text from model weights, but still require a real dataset for training, and match gradients of real data to synthetic data. Concurrent work (Huang et al., 2025) also infers some information from language model weights, focusing on the problem of property inference. Learning from the difference vector between a base and finetuned model has also been recently explored by concurrent work (Lin et al., 2025).

# 3 Approximating Training Data

## 3.1 Threat Model

We consider the increasingly-common scenario of language models that are *open-weights*, but not *open-data*. In particular, we assume access to not just the model weights but a program to run the model as well as optimization details such as the optimization method used and number of training steps. Notably, we also assume access to an initial set of model parameters corresponding to the model state before finetuning, as well as knowledge of the optimizer used (e.g. SGD vs. Adam). This threat model corresponds to many popular LLM releases such as LLAMA-3 (Meta-AI, 2024) and DeepSeek-R1 (DeepSeek-AI, 2025).

Given this small amount of prior knowledge, we seek to approximate the training data used for finetuning. Specifically, we aim to find data that is as effective for finetuning as the actual data used to create the released model.

## 3.2 Background

Formally, we assume access to some training algorithm $\mathcal{T}$ that solves an optimization problem:

$$\theta = \mathcal{T}(\mathcal{L}, \mathcal{D}) = \arg\min_{\theta} \mathbb{E}_{x \sim \mathcal{D}} \left[ \mathcal{L}(x, \theta) \right].$$

Given a loss function $\mathcal{L}$ and a dataset $\mathcal{D}$, the training algorithm produces the set of parameters that minimizes the average loss over datapoints $x \sim \mathcal{D}$.

### 3.3 Optimization Difficulties

To use any technique that requires gradients, we must compute the gradient of the model loss; to compute the model loss, we need labeled data. This is problematic because our threat model assumes access to $\theta_0$ and $\mathcal{L}$, but not $\mathcal{D}$. Given the finetuned model parameters $\theta_f$, we can express our problem as the search for a dataset that, after training, produces a model close to the finetuned model:

$$\mathcal{D}^* = \arg\min_{\mathcal{D}^*} ||\theta_f - \mathcal{T}(\mathcal{L}, \theta_0, \mathcal{D})|| = \arg\min_{\mathcal{D}^*} ||\theta_f - \arg\min_{\theta} \mathbb{E}_{x \sim \mathcal{D}} \left[ \mathcal{L}(x, \theta) \right] ||.$$

Solving this optimization problem has a number of difficulties. First, its nested nature requires training a model on any candidate dataset to obtain a candidate set of parameters $\theta$ that can be compared to the known $\theta_f$. This bi-level optimization is typically expensive but shown to be tractable via brute-force in the dataset distillation literature (Cazenavette et al., 2022).

Second, in case of language models, the dataset $\mathcal{D} = \{(x^{(0)}, y^{(0)}), ..., (x^{(t)}, y^{(t)})\}$ consists of input samples $x^* \in \mathbb{V}^{(s)}$ that are discrete sequences of length $s$ in vocabulary $\mathbb{V}$. Unfortunately, computing the loss $\mathcal{L}(x, \theta)$ requires a non-differentiable lookup operation to convert a token sequence to a sequence of dense embedding vectors. This non-differentiability means that typical dataset distillation approaches are no longer applicable, because we cannot backpropagate directly from $\mathcal{L}$ to $\mathcal{D}$.

## 4 Method: SELECT

Unlike in the differentiable vision case (Wang et al., 2020) we cannot generate data that directly matches the training trajectory of the final model in parameter space. We constrain our problem to data selection instead of data generation: given a large corpus of text data, we search for a small set of datapoints that, after training, produce a model close to the final model.

We hypothesize that the initial update in stochastic gradient descent, computed from the initial model $\theta_0$ should be aligned with the direction leading toward the final model $\theta_f$. Solving this problem requires access to the set of initial model parameters, which we term $\theta_0$. We can express this goal as a search for data $x$ with a gradient that maximizes its projection onto the *model diff* $\theta_f - \theta_0$:

$$\arg\max_{x \in \mathcal{D}} \left[ \nabla \ell(x; \theta_0) \cdot (\theta_f - \theta_0) \right].$$

Since we have access to $\theta_0$, we can compute example-level gradients for all $x \in \mathcal{D}$. A naive solution to this problem might be to take the examples with the top similarity with the parameter difference. However, in practice, this yields highly redundant samples, as it neglects to account for batch-level interactions; when training with stochastic gradient descent, we typically take steps using gradients summed across multiple examples.

In light of this information, we instead express our search as for the set of points that produces a *total gradient* pointing in the direction of the parameter difference:

$$\arg\max_{\mathcal{B} \subseteq \mathcal{D}} \left[ \sum_{x \in \mathcal{B}} \nabla \ell(x; \theta_0) \cdot (\theta_f - \theta_0) \right].$$

Solving for $\mathcal{B}$ exactly requires enumerating all possible subsets of $\mathcal{D}$ and is generally intractable to solve in polynomial time. However, the batch search objective is submodular because it exhibits the diminishing returns property: the marginal gain of adding a new datapoint decreases as the batch grows. The submodularity is known to have an efficient, close-to-optimal greedy solution (Nemhauser et al., 1978).

### 4.1 Creating synthetic checkpoints via linear interpolation

Our threat model only allows us access to a single final model checkpoint, as well as the base model initialization $\theta_0$. State-of-the-art dataset distillation approaches (Cazenavette et al., 2022; Guo et al., 2024) achieve more effective distillation with gradients that match trajectories of *several* final model checkpoints $\theta_j, j \in [1, P]$. This puts us at a significant disadvantage because examples' gradients at the beginning of training may point in a different direction later on during the optimization process. To make up for our lack of additional model checkpoints, we create *synthetic checkpoints* by linearly interpolating between the initial and final model:

$$\hat{\theta}_j = (\frac{j}{P}\theta_0) + (1 - \frac{j}{P}\theta_f),$$

where $P$ is the desired number of synthetic checkpoints. We then search for the batch of examples with a gradient that is most aligned, on average, with the direction of the synthetic checkpoints:

$$\underset{\mathcal{B} \subseteq \mathcal{D}}{\arg\max} \left[ \sum_{j=1}^{P} \sum_{x \in \mathcal{B}} \nabla \ell(x; \hat{\theta}_j) \cdot (\theta_f - \hat{\theta}_j) \right].$$

### 4.2 Efficient per-example gradient computation

Prior work has demonstrated that the gradient of the last layer of language model can be high-resolution enough for synthetic data generation (Nguyen et al., 2025). Since our approach requires per-example gradients, which are typically computationally expensive (Li et al., 2022), we run backpropagation only for the last layer to save memory and reduce overall computation. Specifically, we run a forward pass with a large batch, and then compute batched per-example gradients for only the last layer using `torch.func.vmap`.[2]

### 4.3 Low dimension gradient computation

A key challenge arises from the potentially large size of the dataset $\mathcal{D}$ and the high dimensionality of the gradient $|\nabla \ell|$. Storing all gradients in their original dimension requires $|\mathcal{D}| \cdot |\nabla \ell|$ parameters, which can quickly become prohibitive. To address this, we leverage the classic Johnson–Lindenstrauss lemma (Johnson & Lindenstrauss, 1984), which guarantees that a set of points in $\mathbb{R}^n$ can be mapped to a lower-dimensional space $\mathbb{R}^k$ (for $k \ll n$) while preserving inner products with high probability.

This approach has been successfully applied in prior work (Engstrom et al., 2024; Xia et al., 2024), and we adopt it here to store and retrieve gradients efficiently. Our final SELECT algorithm, which incorporates these projected gradients for batch construction, is summarized in Algorithm 1.

### 4.4 Distilling labels from the final model

Our threat model assumes no access to the true training data. When used for classification, our method produces a synthetic training dataset by labeling a large amount of web text:

$$\hat{y}_i = \arg\max \ell(x_i; \theta_f).$$

Practically, most texts $x_i$ are not relevant to the typical text classification task. We expect that most data will be discarded during selection, and our gradient-based selection will help identify the small subset of points that are actually useful for training.

---

**Algorithm 1** SELECT: Greedy Dataset Creation with Autolabeling

---

**Require:** $\theta, \theta_f, \eta, X, M, d$

1: $\hat{Y} \leftarrow \text{Autolabel}(X, \theta_f)$
2: $\{\hat{\theta}_i\}_{i=1}^N \leftarrow \text{GetSyntheticModelCheckpoints}(\theta, \theta_f)$
3: $G \leftarrow \text{ComputePerExampleGradients}(X, \hat{Y}, \{\hat{\theta}_i\}_{i=1}^N)$
4: $\hat{G} \leftarrow \text{JohnsonLindenstrauss}(G, d)$
5: $g_b \leftarrow \varnothing$
6: $\mathcal{I} \leftarrow []$
7: **while** $|\mathcal{I}| < M$ **do**
8: $\quad i^* \leftarrow \arg\max(\hat{G} \cdot \hat{\theta}_t^T)$
9: $\quad \mathcal{I}.\text{append}(i^*)$
10: $\quad \hat{G} \leftarrow \hat{G} + \text{broadcast}(\hat{G}_{i^*})$
11: **end while**
12: **return** $\mathcal{I}$

---

## 5 Experiments

**Datasets and tasks.** We test models on two tasks, classification and token-level supervised finetuning (SFT). For classification, we train final models on AG News (Zhang et al., 2016), a four-way text classification task, as well as DBPedia (Auer et al., 2007), a fourteen-way classification task, and the 20-Newsgroup dataset (Mitchell, 1997). For SFT, we finetune models on generic web documents sampled from MSMARCO (Bajaj et al., 2018).

For all final model tasks we subsample 10K representative points. We truncate sequences to a maximum of 64 tokens. For classification we use a typical instruction-tuning setup with an instruction token delimiter and all non-label tokens masked. For methods that require seed data, we evaluate using Wikipedia documents drawn from Natural Questions (Kwiatkowski et al., 2019).

**Final models and hyperparameters.** For classification tasks we use GPT-2 medium (Radford et al., 2019); for supervised finetuning we choose the 1B and 3B parameter sizes of LLAMA-3.2 (Meta-AI, 2024). We train final models with the Adam optimizer (Kingma & Ba, 2017) for three epochs with a learning rate of $10^{-4}$. All results are averaged over three seeds, which each produce different final models and seed set splits. To evaluate the performance of a selected dataset, we retrain the model on the selected dataset with the same hyperparameters for 100 epochs and take the best model according to a validation set.

**Baselines.** Since no validation data is available, we do not consider traditional coreset selection methods such as herding. Our main baseline is simple random selection, where data are automatically labeled using the final model and selected at random.

Next, we consider selecting data by gradient features without our greedy gradient-replacement strategy (top-k). Since top-k is prone to selecting highly similar data and our problem space is constrained to balanced classification, we further consider a "balanced top-k" approach, where an adversary selects data based on closest gradient distance but constraints selection to include an equal number of examples from each class.

Although we cannot apply the traditional influence function formulation, we can still compute the perplexity of the seed data and use it as a selection mechanism. We consider the highest- and lowest-likelihood data as baselines (P-Min and P-Max), which is known to be a surprisingly effective baseline for data selection (Yin & Rush, 2025).

We also evaluate a variant of our greedy algorithm, SELECT (batch), which greedily selects examples based on the sum of the gradient within a given batch.

---

[2]https://pytorch.org/docs/stable/generated/torch.func.vmap.html

Table 1: Classification: Main results comparing SELECT on dataset recovery and model performance, with 1K datapoints selected. Seed sets are Wikipedia passages from Natural Questions automatically labeled with the victim model, which is GPT-2 medium (355M parameters).

| | AG News [4 classes] | | | | DBPedia [14 classes] | | | | IMDB [2 classes] | | | |
|---|---|---|---|---|---|---|---|---|---|---|---|---|
| | Vocab ↑ | OTD ↓ | Acc ↑ | Loss ↓ | Vocab | OTD | Acc | Loss | Vocab | OTD | Acc | Loss |
| Random | 82.31 | 1.11 | 65.62 | 0.91 | 73.26 | 1.03 | 40.15 | 2.15 | 75.79 | 1.16 | 55.21 | 0.80 |
| Top-K (Bal.) | 82.85 | 1.10 | 61.15 | 0.97 | 73.04 | 1.02 | 40.04 | 2.01 | 74.41 | 1.17 | 57.71 | 0.69 |
| Top-K | 82.64 | 1.11 | 47.97 | 1.78 | 72.34 | 1.05 | 20.48 | 3.86 | 74.79 | 1.17 | 53.19 | 2.30 |
| P-Min | 83.69 | 1.06 | 63.04 | 0.69 | 63.04 | 1.01 | 53.04 | 1.87 | 54.39 | 1.15 | 54.39 | 0.99 |
| P-Max | 35.51 | 1.12 | 35.51 | 2.07 | 35.38 | 1.03 | 35.38 | 2.70 | 51.25 | 1.15 | 51.25 | 1.75 |
| SELECT-Batch | 84.63 | 1.08 | 75.57 | 0.71 | 76.53 | 0.99 | 58.04 | 1.54 | 77.09 | 1.14 | 59.94 | 0.69 |
| SELECT | **85.01** | **1.07** | **80.05** | 0.70 | **77.67** | **0.98** | 59.91 | 1.54 | **77.99** | **1.13** | **63.79** | 0.71 |
| Expert | | | 88.32 | 0.40 | | | 88.01 | 0.42 | | | 80.87 | 0.51 |

Table 2: SFT: Comparison of selection strategies on dataset recovery and model performance across Llama-3.2 models with 1K datapoints selected.

| | Llama-3.2-3B | | | Llama-3.2-1B | | |
|---|---|---|---|---|---|---|
| | **PPL** ↓ | OT ↓ | Vocab ↑ | **PPL** | OT | Vocab |
| P-Min | 2.76 | 1.331 | 85.0 | 2.63 | 1.330 | 85.1 |
| P-Max | 36.01 | 1.330 | 85.1 | 46.26 | 1.331 | 85.0 |
| Random | 3.31 | 1.331 | 85.1 | 4.15 | 1.331 | 85.1 |
| Top-K | 9.93 | 1.329 | 85.1 | 20.97 | 1.331 | 85.2 |
| SELECT (Batch) | 2.67 | 1.330 | 85.0 | 3.44 | 1.330 | 85.3 |
| SELECT | **2.30** | 1.329 | 85.0 | **2.61** | 1.329 | 85.2 |
| Expert | 2.01 | | | 2.15 | | |

**Metrics (Lexical).** To evaluate lexical similarity, we compute the 'containment similarity' at the vocabulary level by computing the union of tokens present from each dataset. This metric is defined as the maximum ratio between the size of the intersection and the size of either token set, capturing the degree of exact and partial token overlap. Lexical metrics have been extensively employed in data extraction and memorization studies (Carlini et al., 2021; 2022), making them robust candidates for evaluating data recovery.

**Metrics (Embedding).** To capture semantic similarity, we evaluate dataset correspondences in embedding space. Specifically, we compute sentence-level embeddings and solve for the optimal transport (Wasserstein) distance between the resulting embedding distributions (Kusner et al., 2015). We employ both the full optimal transport formulation and, if too expensive, its efficient entropic regularization variant via the Sinkhorn algorithm (Cuturi, 2013). These embedding-based metrics capture semantic correspondences that are otherwise not be captured through lexical measures.

# 6 Results

**Main results.** We first compare our methods with a fixed seed set size of $1k$ selected from $10k$ points; results are shown in Table 1. The Top-K gradient-based data selection performs the worst overall, with the lowest evaluation accuracy after finetuning and relatively high optimal-transport distance. We see significant improvement by simply balancing the top-k choices by class, but this method still underperforms random selection on all three datasets. Random selection is by definition near-balanced for any balanced dataset. SELECT, on the other hand, naturally balances the class distribution of selected datapoints through gradient comparison.

In general we come close to the performance of the final model (10K sample dataset) on AG News and lag behind on DBPedia and IMDB, which have more classes and may be more difficult to compress into fewer samples (since we are selecting 1000 from $10,000$ datapoints).

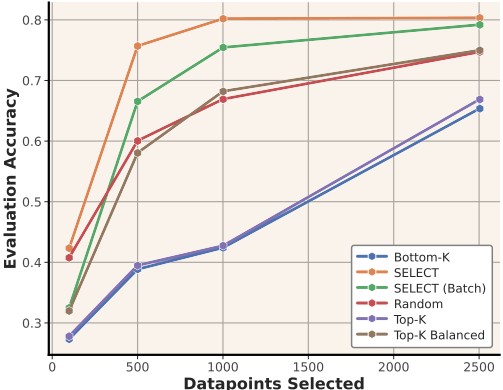 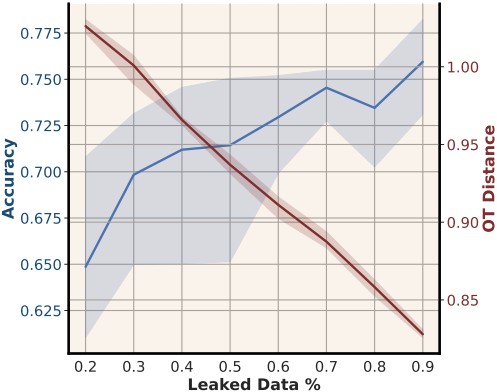

Figure 2: Data selection performance on AG News while scaling amount of selected data from 100 to 2.5K (selected from a seed set of size 10*k*).

Figure 3: Accuracy (y-axis, left) and OT distance (y-axis, right) vs the percent of seed data that is leaked from the true dataset (x-axis).

In our SFT experiments (Table 2), we are interested in examining whether per-example gradients might be useful for data selection. We find that SELECT is significantly effective for selecting data for SFT on both the 1B and 3B-parameter variants of LLAMA, giving better perplexity than any baselines. All methods perform similarly in recovery as measured by OT and Vocab, indicating that the seed set (MSMARCO) is already highly similar to the true finetuning set (Wikipedia documents from Natural Questions).

**Scaling selection size**   In previous experiments we only considered selecting 10K datapoints. We then examine the performance of various methods as the number of datapoints selected scales from 100 to 2.5K. We see SELECT achieves similar performance to random sampling with just 100 samples and significantly outperforms it at higher numbers of samples. In all cases, selecting data by simple Top-K gradients performs poorly. We hypothesize that this happens because the datapoints with the highest gradient similarity to the parameter difference are similar to one another, causing a highly similar subset to be selected, which is not useful for finetuning.

**Analyzing the affect of leakage.**   Finally we consider the case where the finetuning data is publicly leaked and available as some subset of our public seed dataset. We titrate the number of additional non-finetuning samples from 90K, where the leaked data is only 10% of the available data, down to 1K, where the leaked data comprises approximately 90% of the publicly available data.

We see in Figure 3 that our method improves with the percentage of the seed dataset that corresponds to leaked data in both accuracy and distance from the original dataset. These results indicate that an adversary may be able to 'locate' true finetuning data when it is exposed as part of a public finetuning dataset.

## 7   Ablations

**Optimal transport metric analysis.**   We perform ablation by evaluating the sensitivity of our dataset similarity metrics to incremental improvements in data recovery. Specifically, we progressively replace the recovered dataset of 1k datapoints with the ground-truth examples, in 10% increments. For each replacement level, we run five trials and report the average. We observe in Figure 4 that both lexical and embedding metrics improve predictably as selected data are replaced by ground-truths.

| | Seed Dataset | | | | | | |
|---|---|---|---|---|---|---|---|
| | AG NEWS | DBP | IMDB | NEWS | RT | MSM | NQ |
| AG NEWS | **81.8** | 50.5 | 54.8 | 69.9 | 36.3 | 74.4 | 79.2 |
| DBPEDIA | 27.5 | **60.3** | 17.5 | 19.4 | 11.3 | 48.9 | 59.4 |
| IMDB | 60.9 | 52.6 | **75.6** | 63.7 | 67.5 | 67.1 | 64.8 |

Table 3: Comparison of datasets when used as seed (horizontal) vs. test (vertical). Base model is GPT-2 Medium; we select 1k with 100k seed documents. Natural Questions is a useful seed, achieving close to the performance of the true data on two of three tasks.

Table 4: Optimizer Performance Comparison: optimizer used to train the ground-truth model (horizontal) vs. the expert model, after selection (vertical).

| | SELECT | | | Random | | |
|---|---|---|---|---|---|---|
| Optimizer | Adam | AdamW | SGD | Adam | AdamW | SGD |
| Adam | 2.15 | 2.44 | 1.97 | 3.42 | 3.42 | 3.42 |
| AdamW | 2.27 | 2.53 | 1.99 | 3.39 | 3.39 | 3.39 |
| SGD | 2.38 | 2.73 | 2.12 | 3.75 | 3.75 | 3.75 |

For the embedding-based metric (optimal transport distance), the distance decreases to 0.685 as recovered data are fully replaced by ground truth. Importantly, even when one dataset is a strict subset of the other, the OT distance remains nonzero because the larger set's mass is distributed over additional examples, incurring extra transport cost under uniform weighting. In contrast, the vocabulary-level containment similarity saturates at 100% once the recovered data are fully replaced by ground truth, showing complete lexical overlap.

**Seed data distribution.** Most of our experiments include only seed data from Wikipedia passages, which have been shown to be effective enough to train news, topic, and sentiment classifiers. We question whether how much results depend on the chosen distribution of seed dataset. We explore a variety of task-seed data distribution pairs. For each pair, we run SELECT to choose 1K representative points from a seed set of 10K, and show all pairwise results in Section 7.

Perhaps unsurprisingly, each dataset is most useful to seed a model for itself, but we do see interesting interactions. The Newsgroup dataset proves useful seed dataset for AG News, and Rotten Tomatoes movie reviews are still helpful for IMDB. Our default seed dataset of Natural Questions is surprisingly useful in each category, although it lags behind using the same distribution.

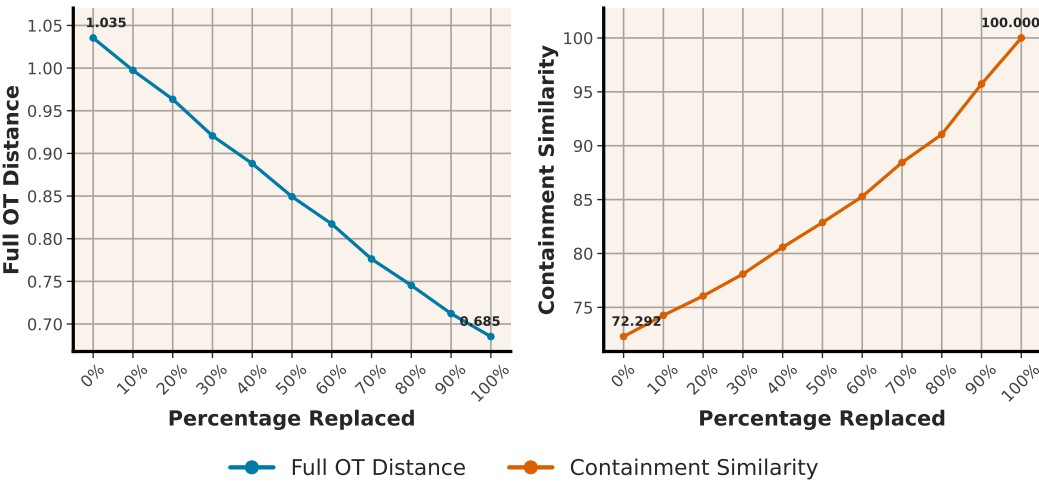

Figure 4: Dataset Evaluation Metrics Ablation.

**Optimizer comparison.** We compare the choice of optimizer used for finetuning the ground-truth model vs. final expert model, to determine whether knowledge of optimizer is a necessary assumption of our threat model. Results are shown in Table 4. Values are test perplexity from SELECT/Random in our the pretraining setting with LLAMA-1B. We observe that using AdamW to train the expert model slightly deteriorates performance on our method, likely due to the weight-level signal lost to weight decay. SGD, which has no second-order optimization, provides even better selection signal than raw Adam. (We plan to add this experiment to the ablation section, space-permitting, in the camera-ready version.)

# 8 Conclusion

We introduce the problem of approximating finetuning data given only model weights, e.g., those available from an open-weights model release. We propose an approach that selects a subset of a large text corpus, then show that the selected subset is sufficient to train a model whose performance is close to the open-weights model. We develop metrics for dataset approximation based on optimal transport of sentence embeddings and show that our method improves over random selection according to these metrics.

These results show that releasing models as open weights, without the corresponding training data, reveals sufficient information to enable selection of effective substitute datasets. Finetuning on these datasets significantly improves performance vs. alternative data selection methods and, in some cases, even comes close to the performance of the original open-weights models.

# Acknowledgments

Research for this project was supported by NSF Grant DRL-2229873, NSF CAREER 2037519, IARPA HIATUS Program 2022-22072200003, and an NSF GRFP.

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

## A   Ethics

The question of recovering training data from the weights of machine learning models is related to protecting the intellectual property of companies that train these models. Model creators often want to release model weights, so that their models might gain users, but do not want to release the data, for many reasons.

Our work is the first step towards recovering information about this unknown data, which may already be possible with the model weights that are public today. Ours is but the first exploration into this problem space, yet our method is able to find similar data available on the Web (which may or may not be the exact training data).

It remains unknown exactly how much information about the training data can be extracted or leaked from the weights of open models. We advise model creators to approach open-weights model releases with caution.

# B  Ablation: Gradient Store

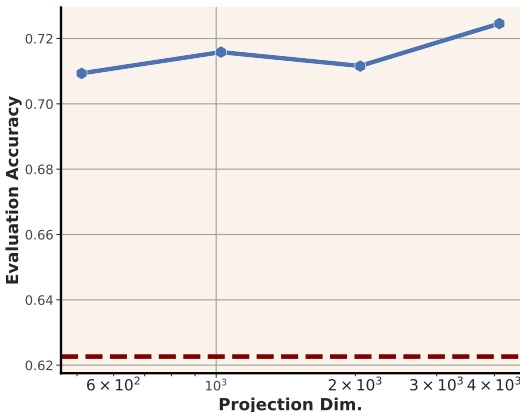

Figure 5: Performance of SELECT (blue) vs. random selection (red, dashed) on selecting 1K points from Natural Questions to train an AG News classifier. We note that our method improves somewhat with a larger projection dimension, although improves over random with a dimension of 512.

**Gradient store dimension.**   We rerun our base experiment (NQ -> AG News, selection size of 1K, seed size of 10K) with a variety of projection dimensions. We compare SELECT across projection dimensions to random selection in Figure 5. We see that gradient-based selection outperforms random even with a dimension as small as 512.[3]

---

[3]In addition to storage size, the total number of FLOPs grows quadratically with the dimensionality of the projection. Thus, we opt to use a size of $4,096$ for all experiments.

