# OpenReview forum: "Approximating Language Model Training Data from Weights"
_colmweb.org/COLM/2025/Conference — COLM 2025_

### Official Review · Reviewer_T5hs · 2025-05-12

**Rating:** 6
**Confidence:** 2
**Ethics Flag:** 1

**Summary:**

This paper introduces a gradient-based data selection method designed to approximate the training data of a fine-tuned language model using only the base and final model checkpoints and a large unlabeled text corpus. By selecting data whose gradients align with the weight difference vector, the authors show that it is possible to reconstruct a dataset that achieves similar downstream performance to the original training data.

**Questions To Authors:**

What happens when the seed dataset is large and diverse (e.g., Common Crawl)? Does your method still recover task-relevant data?

**Reasons To Accept:**

1. The method is technically interesting and makes clever use of weight-space geometry and gradient submodularity.
2. Good ablations. The authors probe various aspects of their proposed method such as seed set distribution, gradient projection dimension,  etc.

**Reasons To Reject:**

1. The paper assumes access to both the base and fine-tuned model weights, the training algorithm, optimizer type, and training steps. In practice, many open models do not release all these details just the weights.
2. The experiments are restricted to small-scale classification tasks and relatively small models. Unclear if the findings will translate to realistic scenarios such as instruction-tuned LMs.
3. Optimal transport on sentence embeddings is used to assess recovery, but the authors do not discuss how well this correlates with actual overlap or usefulness of recovered data. It’s possible to achieve good OT scores without semantically meaningful overlap.
4. The comparison to “random” and “top-k gradient” baselines seems insufficient. More sophisticated baselines from data distillation or data selection literature (e.g., influence functions) are not considered. I understand they are slow, but the experimental setup considered in this paper will work just fine with influence functions.

---

> ### Author Response · Authors · 2025-06-02
>
> Thanks for the suggestions. We have changed our setup to mimic instruction tuning, added a new task (SFT), and scaled to larger models. Full experimental results are available in the general response, but we provide a point-by-point rebuttal below:
>
> > The paper assumes access to both the base and fine-tuned model weights, the training algorithm, optimizer type, and training steps. In practice, many open models do not release all these details just the weights.
>
> This is a fair criticism. We do provide empirical data in our response to Reviewer N6vu showing our method works reasonably well regardless of optimizer choice. And our method does not require access to the true amount of training steps. We do require both sets of model weights and knowledge of the training task, which is not always the case, although it is somewhat common (e.g. the LLAMA base and instruction-tuned models, or DeepSeek base vs. R1).
>
> > The experiments are restricted to small-scale classification tasks and relatively small models. Unclear if the findings will translate to realistic scenarios such as instruction-tuned LMs.
>
> We have rewritten the main experiments to use an setup closer to instruction tuning for classification tasks, as well as larger models for a pretraining data selection task. Our results seem to scale to new tasks as well as 10x larger models. Please see the general response for empirical data.
>
> > Optimal transport on sentence embeddings is used to assess recovery, but the authors do not discuss how well this correlates with actual overlap or usefulness of recovered data. It’s possible to achieve good OT scores without semantically meaningful overlap.
>
>
> In section 6, our analysis shows precisely this: that increasing the percentage of leaked data (i.e., overlap of recovered data) decreases our OT scores, whereas task accuracy increases (see Figure 3).
>
> Importantly, it is not possible to obtain good OT scores without semantically meaningful overlap, because the **optimal transport distance is computed between sentence embeddings**.
>
> For example, given two sets of sentence embeddings ${e(x_i)}^n_{i=1}$ and ${e(y_j)}^m_{j=1}$ and their respective empirical distributions $\mu$ and $\nu$, the OT score (Wasserstein-1 distance) can be calculated as:
>
> $$
> W_{1}(\mu, \nu)
> \;=\;
> \min_{\pi \in \Pi(\mu, \nu)}
> \sum_{i=1}^n \sum_{j=1}^m \pi_{ij} \left\lVert e(x_i) - e(y_j) \right\rVert_{2}
> $$
>
>
> The OT score is fundamentally an embedding metric: If embeddings for A and B are far apart, OT will register a large cost, and vice versa.
>
> > The comparison to “random” and “top-k gradient” baselines seems insufficient. More sophisticated baselines from data distillation or data selection literature (e.g., influence functions) are not considered. I understand they are slow, but the experimental setup considered in this paper will work just fine with influence functions.
>
> Importantly, most data-selection methods--including influence function formulation--rely on reference/validation data to perform selection (Pruthi et al., 2020; Xia et al., 2024).
>
> Specifically, they compute:
>
>
> $\nabla \ell\bigl(w_{t_i},\,z\bigr)\;\cdot\;\nabla \ell\bigl(w_{t_i},\,z'\bigr)$
>
> wher $z$ is a training example and $z'$ is the reference/validation example.
>
> In our setup, we only have access to model weights but not reference data.  The only other baselines that don’t depend on reference data use perplexity (i.e., selecting examples based on the model’s loss). We have included additional perplexity-based baselines to approximate training data in the general response (please refer to the table in the main response).
>
> *Reference:*
>
> Pruthi, Garima, et al. "Estimating training data influence by tracing gradient descent." Advances in Neural Information Processing Systems 33 (2020): 19920-19930.
>
> Xia, Mengzhou, et al. "Less: Selecting influential data for targeted instruction tuning." arXiv preprint arXiv:2402.04333 (2024).
>
>
> > What happens when the seed dataset is large and diverse (e.g., Common Crawl)? Does your method still recover task-relevant data?
>
> We haven't evaluated with exactly Common Crawl due to the large computational requirement of computing per-example gradients for the entire web. But we did evaluate with data from the MS Marco corpus, which also includes randomly sampled documents from webpages, similar to a typical Common Crawl distribution. We still observe that our method is able to find task-relevant data when using MS Marco as seed data.

---

> > ### Comment · Reviewer_T5hs · 2025-06-08
> >
> > Thank you for your detailed response, especially for including the new results.
> >
> > Could you provide more details of the SFT setup? Why did you train on MS MARCO and not a typical instruction tuning scenario?

---

> > > ### Author Response · Authors · 2025-06-08
> > >
> > > We chose MS MARCO as the dataset closest to a typical pretraining dataset. The new results for the *classification* task are computed under an instruction tuning setup, with responses formatted as e.g. `<INST>World News</INST>`, and loss only computed across response tokens.

---

> > > > ### Author Response · Authors · 2025-06-10
> > > >
> > > > Dear Reviewer T5hs,
> > > >
> > > > We are following up as today is the last day for author-reviewer discussion. Please let us know if you need any further clarifications, on our new experimental setup or anything else. Thanks.

---

### Official Review · Reviewer_pFEW · 2025-05-12

**Rating:** 6
**Confidence:** 3
**Ethics Flag:** 1

**Summary:**

The authors propose a gradient-based approach for approximating private training data of publicly available model weights. Their method "SELECT" relies on having a base and fine-tuned open weight models as well as an unsupervised seed dataset. The authors use the "final model" checkpoint to assign labels to data instances, then compute synthetic checkpoints, use the Johnson-Lindenstrauss lemma to compute low-dimensional gradients, and finally perform greedy data selection by picking instances with the highest running gradient sum.

The authors evaluate their method on GPT-2 Medium and Tiny-Llama, and evaluate their methods on three different classification datasets comparing it against Random selection and Top-K selection. The authors, restrict their method to the last layer of the model in order to save compute. They also perform Ablations using optimal transport metric analysis.

**Questions To Authors:**

- How does your method scale to larger models and datasets? While this might not be easy to answer with your current compute resources, providing an estimate based on your experiments would be useful.
- How does your method compared both in performance and computational needs to more traditional methods for data selection? Especially for pre-training.
- Figure 2 is completely unreadable for a colorblind person. Please use a more accesible palette. Most libraries today have them and around 8% of the male population suffers from colorblindness so this will make your paper more accessible.
- Please refrain from using notation like $\theta^T$, this is already standard notation for something else.

**Reasons To Accept:**

- The paper is clear and well written.
- The topic and experiments seem highly relevant to some of the most important current discussion regarding LLMs and copyrighted data.
- Even though the models used are small, the authors show the effectiveness of their methods and perform ablations.
- The authors plan to release the code along with instructions fro reproduction.

**Reasons To Reject:**

- While using the last layer of language model can be high-resolution enough for synthetic data generation, I don't immediately see how this translates to your method.
- At least from the way the method is presented, it seems to rely a lot on the seed selection, so if the proposed method wants to be used to approximate training data, this seems like a major challenge as most of the datasets for current LLMs might not even be publicly available, so your method is not applicable.
- Even though you say that even in the cases when the training dataset is not available, your method also works for data selection, there is a lack of comparison between your method an more traditional techniques for data selection.
- There is an important discussion missing in this paper of how the method scales computationally as the model size and dataset sizes increase, this is not evident from the paper as the models and datasets used were rather small compared to modern standards.

---

> ### Author Response · Authors · 2025-06-02
>
> Hi! Thanks for the helpful review. We provide a response to each of your suggestions below:
>
> > While using the last layer of language model can be high-resolution enough for synthetic data generation, I don't immediately see how this translates to your method.
>
> Recall our formulation of the method in section 4:
>
> $\arg\max_{x \in \mathcal{D}} \Bigl[\,\nabla \ell\bigl(x; \theta\bigr)\;\cdot\;\bigl(\theta_{f} - \theta\bigr)\Bigr].$
>
> Computing the above equation requires taking inner product of the gradients with respect to the training data $x$ and model weight’s difference $\theta_{f} - \theta$.
>
> However, directly using such high-dimensional gradient vectors as features for dataset selection is very computationally expensive, so we use only the last layer alongside random projection to construct meaningful low-dimensional gradient features.
>
>
> > At least from the way the method is presented, it seems to rely a lot on the seed selection, so if the proposed method wants to be used to approximate training data, this seems like a major challenge as most of the datasets for current LLMs might not even be publicly available, so your method is not applicable.
>
>
> Indeed, seed selection is crucial, as seed datasets must be close to the ground-truth training data. Lacking access to the true training data is a challenge not only for our method but for the entire field of training-data recovery.
>
> However, our method remains applicable even when ground-truth training data is not publicly available. In our experiments, we are able to identify lexically and semantically relevant data and recover performance comparable to that of the final model weight.
>
>
> > Even though you say that even in the cases when the training dataset is not available, your method also works for data selection, there is a lack of comparison between your method an more traditional techniques for data selection.
>
> Importantly, traditional data selection relies on reference/validation data to perform data selection: lexical methods or embedding-based methods select training data based on lexical or semantic similarity with reference data [Albalak et al., 2024].
>
> In our problem set up, we only have access to model weights but not reference data.  The other baselines that didn’t depend on is perplexity data selection, which select data based on model’s loss. We included additional baselines for using model’s perplexity to approximate training data in the table in the main response.
>
>
> Reference:
>
> Albalak, Alon, et al. "A survey on data selection for language models." arXiv preprint arXiv:2402.16827 (2024).
>
> > There is an important discussion missing in this paper of how the method scales computationally as the model size and dataset sizes increase, this is not evident from the paper as the models and datasets used were rather small compared to modern standards.
>
> We agree. We have since provided additional experiment on instruction-tuning on a larger 1B language model in the main results. Our method seems to scale well to 10x larger models, as well as a more modern task setup of selecting pretraining data.
>
> > How does your method scale to larger models?
>
> We have new experiments with larger models in the general response, please let us know if you have any questions.
>
> >  Figure 2 is completely unreadable for a colorblind person. ... Please refrain from using notation like \theta^T ...
>
> Thanks for these small suggestions. We will fix the figure style and change the superscript to subscript in the final version (as we can't update the PDF during this rebuttal period).

---

> > ### Comment · Reviewer_pFEW · 2025-06-09
> >
> > > Recall our formulation of the method in section 4:
> > >
> > > $\arg\max_{x \in \mathcal{D}} \Bigl[,\nabla \ell\bigl(x; \theta\bigr);\cdot;\bigl(\theta_{f} - \theta\bigr)\Bigr].$
> > >
> > > Computing the above equation requires taking inner product of the gradients with respect to the training data $x$ and model weight’s difference $\theta_{f} - \theta$.
> > >
> > > However, directly using such high-dimensional gradient vectors as features for dataset selection is very computationally expensive, so we use only the last layer alongside random projection to construct meaningful low-dimensional gradient features.
> >
> > Yes I understand this, and I understand the computation of the high-dimensional gradient vectors is expensive, what I don't see immediately is the evidence to support that the low-dimensional gradient features (that is useful for for synthetic data generation) is also useful for training.
> >
> > > Indeed, seed selection is crucial, as seed datasets must be close to the ground-truth training data. Lacking access to the true training data is a challenge not only for our method but for the entire field of training-data recovery.
> > >
> > > However, our method remains applicable even when ground-truth training data is not publicly available. In our experiments, we are able to identify lexically and semantically relevant data and recover performance comparable to that of the final model weight.
> >
> > I agree with this, but the problem is that the paper is literally called _"Approximating Language Model Training Data from Weights"_, so I think to make this point stronger, the presentation needs to be updated, putting more emphasis on the usability of your method to produce usable training data rather from an existing model, rather than approximating the original training data of the model.
> >
> > > Importantly, traditional data selection relies on reference/validation data to perform data selection: lexical methods or embedding-based methods select training data based on lexical or semantic similarity with reference data [Albalak et al., 2024].
> > >
> > > In our problem set up, we only have access to model weights but not reference data. The other baselines that didn’t depend on is perplexity data selection, which select data based on model’s loss. We included additional baselines for using model’s perplexity to approximate training data in the table in the main response.
> > >
> > > Reference:
> > >
> > > Albalak, Alon, et al. "A survey on data selection for language models." arXiv preprint arXiv:2402.16827 (2024).
> >
> > I don't immediately see how the additional experiments answer the question, an experiment where you directly compare a model trained only with data generated with your method against a model trained with a traditional data selection method would be more useful
> >
> > > We agree. We have since provided additional experiment on instruction-tuning on a larger 1B language model in the main results. Our method seems to scale well to 10x larger models, as well as a more modern task setup of selecting pre-training data.
> >
> > Thank you for the additional experiments, this addresses my concern. And also thank you for making your paper more accessible.
> >
> > This paper presents very interesting ideas. However, I still think there are some concerns to be addressed so I have decided to keep my score as is.

---

> > > ### Author Response · Authors · 2025-06-10
> > >
> > > >  I understand the computation of the high-dimensional gradient vectors is expensive, what I don't see immediately is the evidence to support that the low-dimensional gradient features (that is useful for for synthetic data generation) is also useful for training.
> > >
> > > It's unclear what your exact qualm is, but it is possible you are wondering why selecting using gradients from a single layer translates to full-model finetuning. We did run an experiment comparing using our data for training only the last layer to full finetuning. We saw that freezing the last layer worsens perplexity from 2.09±0.07 to 2.26±0.11, indicating that the expressivity of full finetuning may be necessary important for optimal performance.
> > >
> > > Does this resolve your issue? If not, please try to elaborate as we may be missing your point.
> > >
> > > > the presentation needs to be updated, putting more emphasis on the usability of your method to produce usable training data rather from an existing model, rather than approximating the original training data of the model
> > >
> > > We are not suggesting that our method is useful for generic data selection, only that it is the best option when no groundtruth data is available. Throughout the experiments we demonstrate how closely our method approximates the true training data, via both OT and lexical metrics. That said, we are open to making small changes to the framing – especially in the introduction.
> > >
> > > > an experiment where you directly compare a model trained only with data generated with your method against a model trained with a traditional data selection method would be more useful
> > >
> > > To be clear: this is impossible. All methods are trained on a seed dataset with no access to ground-truth data, so we cannot test traditional dataset selection methods such as lexical, embedding, or even influence functions – all of these require validation data (which we don't have).

---

### Official Review · Reviewer_N6vu · 2025-05-13

**Rating:** 8
**Confidence:** 3
**Ethics Flag:** 1

**Summary:**

This paper addresses an interesting problem by proposing a method called SELECT, which identifies samples from a publicly available dataset that closely resemble the original training set of an open-weight model—despite the original training data being unavailable. The method leverages two versions of the same model: a base model and its fine-tuned counterpart. It uses the base model to compute last-layer gradients and the fine-tuned model to generate pseudo-labels for the samples. Overall, the paper presents a modern take on the coreset selection problem in scenarios where no information about the original training set is accessible.

**Questions To Authors:**

Here are few questions to the authors regarding the design choices for the select algorithm.

1) Why computing gradients is neccessary and can't a similar result be shown with just computing loss which is much cheaper. Is it because this method is inspired by Less (by Xia et. al)  and they use gradients instead of loss.

2) For gradients computation the base model is used and for pseudo labelling the finetuned model is used. Can we use the finetuned model for both?

**Reasons To Accept:**

1) This paper is very well written and has the core message is well written.

2) The design of the method is very well explained and based on prior works (also make sense).

3) This is a hard problem and the authors have used many approximations regarding gradient computation and solving the original bi-level problem.

4) The results overall looks reasonable (although I have some reservations about baselines).

5) The problem is very interesting and very useful for the community.

**Reasons To Reject:**

1) The paper started with absense of training data and availability of model weights but then claims to be a good coreset selection algorithm which is orthogonal to the main story. Refer page 2 para 1 "disjoint from true training set ..."

2) Based on my understanding AdamW is widely used for LLM training not sure why the authors are using adam.

3) The baselines are quite weak -- for instance show us first what are the open weight model performance on the test set of the datasets you are using. Then claim that even without having the train set we have achieved a certain level of performance. This method is beating random subset selection baseline so, it has some merit already.

4) Can you use Openwebtext or any pre-train data and use your method and show how much perplexity difference it has with the open weight base model. As all the results are shown in smaller scale data. Openwebtext is also a small size data should fit in academic compute budget. This will help us gauge the method's impact on pre-training.

---

> ### Author Response · Authors · 2025-06-02
>
> Thanks for the review and helpful comments. We performed a new optimizer ablation showing that our assumption of optimizer used is not necessary; we also added a new task (SFT) which is close to pretraining data selection. Please see our point-by-point response below for more explanation.
>
> > The paper started with absense of training data and availability of model weights but then claims to be a good coreset selection algorithm which is orthogonal to the main story. Refer page 2 para 1 "disjoint from true training set ...”
>
> To clarify, our method performs coreset selection on a seed dataset that is distinct from ground truth training data. We assume no information about the ground-truth training data: in the best case, the training data is a subset of the seed data; in the worst case, the two datasets are disjoint. Regardless, our goal is to select from the seed datasets to reach same model performance as final model checkpoint.
>
> > Based on my understanding AdamW is widely used for LLM training not sure why the authors are using adam.
>
> Thanks for the insightful question. We ran a new ablation comparing the optimizer used to train the ground-truth model (horizontal) vs. the expert model, after selection (vertical). We show results for SELECT (top) and our random baseline (bottom).
>
> #### Data selected by SELECT
>
> |    |   Adam |   AdamW |   SGD |
> |:-----------------|-------:|--------:|------:|
> | Adam             |   2.15 |    2.44 |  1.97 |
> | AdamW            |   2.27 |    2.53 |  1.99 |
> | SGD              |   2.38 |    2.73 |  2.12 |
>
> #### Data selected randomly
>
> |    |   Adam |   AdamW |   SGD |
> |:-----------------|-------:|--------:|------:|
> | Adam             |   3.42 |    3.42 |  3.42 |
> | AdamW            |   3.39 |    3.39 |  3.39 |
> | SGD              |   3.75 |    3.75 |  3.75 |
>
> Numbers are test perplexity from SELECT in our new pretraining setting with LLAMA-1B. We observe that using AdamW to train the expert model slightly deteriorates performance on our method, likely due to the weight-level signal lost to weight decay. The default SGD, which has no momentum term, provides even better selection signal than raw Adam. (We plan to add this experiment to the ablation section, space-permitting, in the camera-ready version.)
>
>
> > The baselines are quite weak -- for instance show us first what are the open weight model performance on the test set of the datasets you are using. Then claim that even without having the train set we have achieved a certain level of performance. This method is beating random subset selection baseline so, it has some merit already.
>
> To clarify, the task is classification: most open-weights models, before finetuning, will perform poorly compared to our random subset selection baselines, which are allowed some finetuning data.  However, we do show the performance of the final model (trained on the *true*, e.g. not extracted, training data) on the test set in the last row of our main experiment (please see Table 1).
>
> We included additional baselines for using model’s perplexity to approximate training data in the table in the main response. We show that SELECT still generally outperforms other baselines, although the max-likelihood method is competitive.
>
> > Can you use Openwebtext or any pre-train data and use your method and show how much perplexity difference it has with the open weight base model. As all the results are shown in smaller scale data. Openwebtext is also a small size data should fit in academic compute budget. This will help us gauge the method's impact on pre-training.
>
> That’s a great observation. We rewrote our codebase to accomodate larger models and alternative (i.e. language modeling) loss and ran experiments on the 1B and 3B LLAMA 3.2 models. While our method is applicable to pre-training, we believe it is somewhat unrealistic to assume access to the weight initialization. Instead, we demonstrate results on supervised fine-tuning (SFT) using the same language modeling loss, on top of the pretrained model. Our method outperforms all baselines on both the 1B and 3B models.
>
>
> > Why computing gradients is neccessary and can't a similar result be shown with just computing loss which is much cheaper.
>
> This was a great suggestion– thanks. We added both highest-loss and lowest-loss selection baselines in both sets of new experiments in the general response.
>
> > For gradients computation the base model is used and for pseudo labelling the finetuned model is used. Can we use the finetuned model for both?
>
> We have tried this but found the gradients of the finetuned model (perhaps surprisingly?) are not useful for the task, and perform worse than random selection. We will provide these results in the appendix of our final paper.
>
> ---
>
> The full set of improved main Table 1 and Table 2 results are available in the general response. Please let us know if you have any questions.

---

> > ### Comment · Reviewer_N6vu · 2025-06-08
> > **Rebuttal Acknowledgement**
> >
> > Hi Authors,
> >
> > Thanks for honoring all my requests and suggestions. It seems the new results has made this paper more concrete. For example (which I pointed out) that loss (likelihood) is a cheap but effective proxy compared to the gradient based selection scheme. As per tables (in general response) it seems it is indeed the case. Please highlight this in the paper's newer version. This intuition of low loss samples (easy samples) in finetuning has been discussed in this work (https://arxiv.org/abs/2502.02797).
> >
> > I am improving my score in light of these new findings.

---

> > > ### Author Response · Authors · 2025-06-08
> > >
> > > Thanks for your help improving our contribution! We will add this citation when we discuss the new baselines in the next version of our paper.

---

### Official Review · Reviewer_cimo · 2025-05-16

**Rating:** 6
**Confidence:** 3
**Ethics Flag:** 1

**Summary:**

This paper presents a method for generating approximating fine-tuning data for a text classification tasks.
- The goal is to find a set of synthetically generated data that can "simulate" the training effect -- measured as either close to the actual fine-tuning data (lexically or semantically) or leading to the same training effect (approximating similar accuracies or losses).
- The key idea of the method is to search for examples that can lead to gradient changes along the direction between the gradient difference between pre-trained model as well as the fine-tuned model.
- Several techniques are considered to improve the efficiency (using only the last layer gradient, plus mapping the stored gradients to lower dimensions) and effectiveness (considering the interpolated weights between the starting and end point) of the algorithm.
- The papers empirically show that the algorithm can outperform several simple baselines, and conducted ablations and analysis on the proposed method to show its effectiveness.

**Questions To Authors:**

- In figure 3, the x axis should be 10, 20.. instead of 0.1, 0.2, .., if I understand it correctly.
- the reference to table 2/3 is not correct (it writes section 7 in the last paragraph on page 8)

**Reasons To Accept:**

- Extracting the training data of the open-weight language models is an interesting setup. It can help people better understand the training dynamics of the language models, as well as the safety of the models.
- The method seems to be reasonable and Novel; the empirical results seem to show the strength of the proposed method.

**Reasons To Reject:**

- The current setup may be a bit synthetic.
    1. First only having the task classification task seems to be a bit limited for such language models -- also the training setup doesn't use the typical instruction fine-tuning setup (as it doesn't use a textual prompt but simply requiring the last token to be a class label).
    2. Furthermore, perhaps what's interesting in this setup is how to avoid some sensitive (exact) training data being recovered; however this method does not have the guarantee for recovering verbatim copies of the training data---all it can do is "approximating" the data---unless it can have access to a good set of seed data. In that spirit, it is closer to dataset distillation, which the authors seem to be aware of. In that case, I suggest the authors change the framing of the paper a bit.
- I think there are several additional experiments for this setup:
    1. As one important baseline, one can select the data based on the logprobs: for example, we can score the sequences using the fine-tuned model, and higher logprobs can indicate more confidence or closer to the training data. Running a top-k experiment using the logprob seems to be an important baseline and I suspect it can achieve nice results.
    2. Since the authors only use the last layer gradient, during the training of the final model, one should also consider freeze all the layers expect for the last one. Also the authors can consider a layer selection method -- identifying the best layers to choose.
    3. An interesting analysis for the experiment in figure 3 is that the authors should report how much of the leaked data is selected using different algorithms.

---

> ### Author Response · Authors · 2025-06-02
>
> Thank you for the constructive feedbacks and and careful reviews. We provide point-to-point response to your concern.
>
> > The current setup may be a bit synthetic.
> >
> > 1. First only having the task classification task seems to be a bit limited for such language models -- also the training setup doesn't use the typical instruction fine-tuning setup (as it doesn't use a textual prompt but simply requiring the last token to be a class label).
>
> Thanks to your suggestion for how to improve our experimental setup: we have adjusted our experiments to use a more conventional instruction tuning setup and scaled to larger models (1B and 3B parameters). Changing the task format slightly improved performance. More importantly, we rewrote our codebase since submission to accommodate arbitrary tasks, which made it simple to test our models on supervised finetuning data selection. We show results in the general response; our method is useful for supervised finetuning. And although the new format makes the baselines better too, SELECT still outperforms all baselines, including max-likelihood data selection.
>
> > 2. Furthermore, perhaps what's interesting in this setup is how to avoid some sensitive (exact) training data being recovered; however this method does not have the guarantee for recovering verbatim copies of the training data---all it can do is "approximating" the data---unless it can have access to a good set of seed data. In that spirit, it is closer to dataset distillation, which the authors seem to be aware of. In that case, I suggest the authors change the framing of the paper a bit.
>
> Thank you for this suggestion. You are right that our method approximates rather than recovers exact training data, and that the connection to dataset distillation is more apt than our current framing suggests. We will revise the introduction to better reflect this distinction and clarify how our approach differs from traditional dataset selection (given the absence of validation data).
>
>
> > I think there are several additional experiments for this setup:
> >
> > 1. As one important baseline, one can select the data based on the logprobs: for example, we can score the sequences using the fine-tuned model, and higher logprobs can indicate more confidence or closer to the training data. Running a top-k experiment using the logprob seems to be an important baseline and I suspect it can achieve nice results.
>
> This is a great point. We added this baseline for all main experiments: for classification, we use the likelihood of the autolabeled class to select data; for SFT, we use the perplexity of the sequence.
>
> In general max-likelihood data selection is a strong baseline, especially for SFT. In some cases its performance comes exceptionally close to that of the gradient-based SELECT (for example, 2.63 vs. 2.61 test perplexity on SFT data selection). See the main response for full results.
>
>
> > 2. Since the authors only use the last layer gradient, during the training of the final model, one should also consider freeze all the layers expect for the last one. Also the authors can consider a layer selection method -- identifying the best layers to choose.
>
> This is true: in some sense, the data is selected to have a useful gradient only for the last layer of the model. We ran a small experiment during this rebuttal to examine the effect of unfreezing the last layer while training the expert model.
>
> In the SFT experiment detailed in the main response, we compared the default setting (all layers unfrozen) to freezing the last layer. We observe that freezing the last layer worsens perplexity from 2.09±0.07 to 2.26±0.11, indicating that the expressivity of full-finetuning may be necessary important for optimal performance.
>
>
> > 3. An interesting analysis for the experiment in figure 3 is that the authors should report how much of the leaked data is selected using different algorithms.
>
> Due to time and compute constraints, we are not able to run this full sweep over data sizes during the rebuttal period. However, we can add this to the final version of the paper (i.e. we can add two additional lines to figure 3: Top-K and random selection).
>
> >  **Questions To Authors:**
> >
> > - In figure 3, the x axis should be 10, 20.. instead of 0.1, 0.2, .., if I understand it correctly.
> > - the reference to table 2/3 is not correct (it writes section 7 in the last paragraph on page 8)
>
> Thank you for the attention to detail. We will fix typos and references in camera ready.

---

> > ### Comment · Reviewer_cimo · 2025-06-09
> > **Thank you for your response!**
> >
> > Thank you for addressing my concerns and adding the experiments! It's helpful to see the SFT results, as well as the likelihood-based baselines. I've increased my score accordingly, in light of the strong baseline results.

---

> > > ### Author Response · Authors · 2025-06-10
> > >
> > > Thank you! As today is the last day for author–reviewer discussion, do let us know if you have additional comments or concerns.

---

### Author Response · Authors · 2025-06-02
**General Response**

### General response: New main results

Thanks to all the reviewers for the helpful feedback, which has significantly improved our draft. We have updated all of our main experiments with the following additions & modifications:

(a) we add the task of token-level language modeling (supervised finetuning / SFT), as well as classification
(b) we scale our method to larger models (1B and 3B parameters, from 300M) and use an instruction-tuning setup for classification
(c) for both the original task (classification) and new task (SFT) we add baselines: choosing highest-likelihood and lowest-likelihood training data – denoted in the table as Likelihood (max) and Likelihood (min).

---

### Classification (GPT-2 355M)


Dataset columns are AG News, DBPedia, IMDB.

| | OTD | Acc | OTD | Acc | OTD | Acc |
|:-----|------:|------:|------:|------:|------:|------:|
| Random |  1.11 | 73.40 |  1.03 | 67.37 |  1.16 | 63.56 |
| Likelihood (max) |  1.12 | 35.51 |  1.03 | 35.38 |  1.15 | 51.25 |
| Likelihood (min) |  1.06 | 83.69 |  1.01 | 63.04 |  1.15 | 54.39 |
| Top-K |  1.10 | 38.20 |  1.05 | 25.50 |  1.16 | 48.83 |
| Top-K (Balanced) |  1.09 | 70.49 |  1.03 | 60.29 |  1.16 | 52.70 |
| SELECT (Batch) |  1.06 | 83.71 |  1.01 | 70.75 |  1.15 | 61.85 |
| SELECT |  1.06 | 82.91 |  1.00 | 70.17 |  1.15 | 62.43 |


---


### Supervised finetuning (LLAMA-3.2 1B and 3B)

| Model | Strategy |  Perplexity | OT |
|-------|----------|----------------------------|---------------------------|
| **LLAMA 3.2 3B** | Top-K | 9.93 | 1.329 |
| | Random | 3.31 | 1.331 |
| | Likelihood (max) | 36.01 | 1.330 |
| | Likelihood (min) | 2.76 | 1.331 |
| | SELECT (Batch) | 2.67 | 1.330 |
| | SELECT | 2.30 | 1.329 |
| **LLAMA 3.2 1B** | Top-K | 20.97 | 1.331 |
| | Random | 4.15 | 1.331 |
| | Likelihood (max) | 46.26 | 1.331 |
| | Likelihood (min) | 2.63 | 1.330 |
| | SELECT (Batch) | 3.44 | 1.330 |
| | SELECT | 2.61 | 1.329 |

For SFT, we train the initial model on documents from the MS Marco. Throughout all experiments, we select the top 1000 points from a seed set of 10,000 documents from Natural Questions. The new min-likelihood baseline is our most effective baseline in several settings. However, SELECT is still the most effective method for data selection across tasks and model sizes.

---

### Author Response · Authors · 2025-06-07
**Rebuttal follow-up**

Hello ACs and reviewers!

Thanks for your reviews, which were already very helpful in improving our work.

We updated our paper with new main results on larger models (from the Llama 3 family), a new task (SFT), and a new format closer to traditional instruction-tuning. We also performed an ablation showing the effect of different optimizers on our data selection algorithm and tested our model when training the last-layer only. And we answered a large number of questions from reviewers, including considering a slightly different framing of our work.

We are aware that the COLM rebuttal period only lasts three more days, until June 10th. We would welcome the opportunity for another round of feedback. Please let us know if you have any further questions.

– Authors of "Approximating Language Model Training Data from Weights"

---

### Decision · Program_Chairs · 2025-07-08

**Decision:**

Accept

**Comment:**

This paper introduces a gradient-based approach to approximate the data a language model has been fine-tuned on.

Reviewers all agree that the proposed approahc tackes an importnat problem, is original, and is supported by good experimental evidence. They also note that the manuscript is well written, and that the proposed method should be easy to reproduce.

Reviewers raised several criticism during discussion period; authors have incorporated nearly all of them, significantly improving the quality of the paper. 2 out of 4 reviewers expressed satisfaction with new experimental results and explanations, raising their score.

**Pros**

- Solid motivation for this work (cimo,N6vu,pFEW)
- Novel method (cimo, N6vu,T5hs)
- Reasonable approach (cimo,N6vu)
- Good experimental results (pFEW,T5hs)
- Should be easy to reproduce (pFEW)
- Well written (N6vu,pFEW)

**Cons**

- the experimental setup is uncommon; instead of instruction-tuning, it considers more narrow classification tasks (cimo, N6vu)
  - fixed in rebuttal
- the proposed method is not extracting data, more similar to distillation (cimo)
- missing several baselines (cimo)
  - fixed in rebuttal
- paper uses adam instead of more common adamw (N6vu)
  - fixed in rebuttal
- Requires access to data similar to training data (pFEW) as well as training algo, optimizer type (T5hs)
- unclear how paper scales (pFEW)
  - partially fixed in rebuttal